# The multisystemic roots of South African child and youth resilience: A scoping review

**Linda C. Theron** [1]*, **Adrian D. van Breda** [2]

**1** Department of Educational Psychology, University of Pretoria, Pretoria, South Africa, **2** Department of Social Work & Community Development, University of Johannesburg, Johannesburg, South Africa

* linda.theron@up.ac.za

## Abstract

### Introduction and objective

A multisystemic approach to researching resilience investigates resources across multiple systems, including biological, psychological, social, institutional, structural, environmental, and cultural systems, with special interest in how these resources co-act to enable better-than-expected outcomes among risk-exposed children and youth. This approach is an important step toward redressing neoliberal misinterpretations of resilience as a personal capacity. However, it is unclear how well a multisystemic approach is reflected in recent studies of child and youth resilience conducted in South Africa, a country where ongoing structural violence demands resilience from most children and youth. In response, this article reports a scoping review of South African child and youth resilience studies published between 2018 and 2023.

### Methodology

The methodology aligned with the PRISMA extension for Scoping Reviews. The authors systematically scoped the available literature ($n = 1309$ records) to determine which resources from which systems were associated with the resilience of South African children and youth (birth to 29 years). Using a multisystem resilience framework, the narrative review of 99 eligible studies documents the biological, psychological, social, institutional, structural, environmental and cultural resources that enabled better-than-expected outcomes among risk-exposed children and youth.

### Results

Psychological and social resources were more prominently reported than biological, institutional, structural, environmental or cultural resources. Two-thirds of the included studies reported resources from two or more systems, with psychological and social systems dominating multisystem studies. Despite the inclusion of multiple systems, studies seldom reported co-acting resources.

**Data availability statement:** All relevant data are within the paper and its Supporting information files. As this is a systematic review of other research, the paper includes the database, search strategies and findings which support replication of the study.

**Funding:** The author(s) received no specific funding for this work.

**Competing interests:** The authors have declared that no competing interests exist.

## Discussion

Although the attention to resources across multiple systems is encouraging, child and youth resilience agendas will be better served by studies that document co-acting resources. This will allow policymakers and service providers to gauge the additive effects of multiple resources and which combinations of resources are most likely to advance young people's resilience.

## Introduction

The study of child and youth resilience first piqued researcher interest in the 1970s. Aided by relentless global risk, including the COVID-19 pandemic and the climate emergency, child and youth resilience continues to dominate research agendas [1]. While definitions vary [2], resilience is commonly understood as the ability to adjust positively to significant risk exposure [3,4]. Over time, however, what this ability is attributed to has changed, with more recent studies identifying multiple resources from multiple systems as integral to child and youth resilience [5]. This move toward using a multisystemic approach to researching and theorizing resilience is an important development, as multisystemic accounts of resilience preclude children and youth being held responsible for their (in)ability to adjust to significant stress, in contrast to earlier person-focused, neoliberal accounts of resilience [2].

South Africa makes an interesting case study for child and youth vulnerability and resilience. Its large percentage of children and youth in the population [6], high levels of child poverty and multidimensional child poverty (which extends poverty beyond mere financial status) [7], exceptionally high youth unemployment [8] and financial inequality [9], and the disturbing levels of gender-based violence [10] in combination provide the case of a country faced with major social, political and economic challenges. In this regard, South Africa is similar to most majority world countries. The majority world, encompassing most of Latin America, Africa, and Asia, is home to 90% of the world's children and youth and characterized by relentless risks [11]. On the other hand, by contrast with many African and other majority world countries, South Africa is stable and developed. In this context, vulnerability and resilience may both be highlighted, providing useful points of comparison with both developing and developed countries.

How well have South African studies of child and youth resilience kept pace with the trend to explain resilience as a multisystemic process? This is a crucial question in the context of South Africa, where young people's exposure to significant risk is high. Most South African children and youth have first-hand experience of poverty, hunger, violence, HIV-related challenges (e.g., being HIV+ or living with a caregiver who is HIV+), inequitable access to services, including quality education, and the climate emergency [12–14]. Girls are disproportionately vulnerable, with high numbers becoming teenage mothers [15] or being exposed to gender-based violence [16]. At least one in three South African youth is not in education, employment or training

(NEET) [17]. Worryingly, systemic support aimed at enabling positive adjustment to the aforementioned risk exposures is typically tepid [13].

Given this reality, there is a very real risk that South African young people could be made solely responsible for their own resilience. To do so would be to magnify the risks they already face, not least because risk is often multisystemic [1]. Signposting that child and youth resilience is rooted in multiple co-acting resources found across multiple systems is one way to prompt policymakers, service providers, communities, and families to be co-accountable for young people's resilience. The scoping review that we report in this article acts as that signpost.

Initially, resilience studies reified personal strengths [18]. Psychological strengths were particularly emphasized, including self-esteem, self-regulation skills, agency and mastery, perseverance or grit, and hopeful meaning-making [3]. While personal strengths have remained important to child and youth resilience, they could not fully account for young people's ability to adjust well to significant risk and so subsequent theories foregrounded the role of resilience-supporting social ecologies (e.g., families, schools, communities) [19,20]. From a multisystemic perspective, a more complete account would need to go beyond the social ecology to acknowledge resources found in biological and environmental systems also [21].

Understood multisystemically, young people's ability to adjust well to significant risk is rooted in co-acting resources from multiple systems, including biological, psychological, social, institutional, structural, environmental, and cultural systems [4,21–24]. Put differently, a multisystemic approach directs attention to multiple system resources found across the micro-macro continuum [1]. These include resources in young people's bodies (e.g., neurological strengths), psyche (e.g., hopefulness), family (e.g., caring parents), immediate community (e.g., supportive neighbors or accessible services), immediate environment (e.g., accessible green spaces), as well as distal ones, such as enabling government policy (e.g., cash transfers to resource-constrained households or structural justice) and enabling cultural heritage (e.g., comforting spiritual or religious beliefs and practices).

Ultimately, a multisystemic account identifies a combination of resources working in tandem to support young people to adjust well to significant risk [5,25]. For example, during the COVID-19 pandemic, Masten and Motti-Stefanidi [26] explained child and youth resilience to pandemic-related challenges as a multifaceted capacity that was co-informed by diverse resources, such as health, immune function, social support, community density, access to healthcare services, adapted school rules and functioning, and comforting spiritual beliefs.

There are glimmers of multisystemic thinking in earlier studies of South African child and youth resilience. A systematic synthesis by Van Breda and Theron [27] of 61 South African child and youth resilience studies conducted between 2009 and 2017 showed that personal and relational resources were well accounted for, along with nascent attention to institutional, structural, and cultural system resources. However, this review made no mention of environmental resources. Likewise, it did not explicate what combinations of resources were reported.

In response to these lacunae and the need to advance multisystemic understandings of child and youth resilience in the risk-saturated context of South Africa, this article updates the Van Breda and Theron [27] review. The updated review answers two broad questions: What resources, as documented in studies published between 2018 and 2023, support the resilience of South African children and youth? And, how aligned are the reported resources with multisystemic perspectives of resilience? The latter question directs attention to the types of systems (i.e., biological, psychological, social, institutional, structural, environmental, and cultural) implicated in the reported resources, as well as to resource interaction (i.e., reports of co-acting resources or resource combinations).

## Method

To conduct the review, we followed the guidance in the PRISMA extension for Scoping Reviews (PRISMA-ScR) [28] (see S1 Checklist).

## Eligibility criteria

Resilience studies have three hallmarks [29]. They investigate (i) a better-than-expected outcome (e.g., educational progress or success, mental health, civic engagement, or successful transitions) in the context of (ii) significant risk (e.g., exposure to or experience of maltreatment, conflict, structural disadvantage, or life-threatening illness) and account for (iii) the promotive and protective factors or processes (PPFPs) that are associated with that better-than-expected outcome. Understood this way, resilience is a process, rather than an outcome. Using these hallmarks [29], we considered publications eligible for inclusion if they reported the PPFPs associated with a better-than-expected outcome among children, adolescents, and emerging adults [18-to-29-year-olds; 30] with experience of significant stress exposure and living in South Africa.

Following Ungar [29], we excluded studies that reported resilience as the outcome. Further, because this scoping review updates Van Breda and Theron [27], we excluded any studies published before 2018. Like Van Breda and Theron [27], we excluded intervention studies when they reported the results of interventions aimed at achieving a better-than-expected outcome (but included them when they reported baseline data or non-intervention findings) and publications that reported no original research results (e.g., commentaries and systematic reviews).

## Information sources and search

An independent research psychologist searched for relevant academic journal publications using multiple databases: Africa-Wide, CINAHL, ERIC, PsycARTICLES, and PsycINFO (all via EBSCOhost platform), Medline (via Web of Science Clarivate Analytics), PubMed, Scopus (which includes contents of Embase), Web of Science Core Collection, and SciELO Citation Index. In the interests of quality, we delimited the search to articles published in peer-reviewed journals. The time parameter for the online or full publication of these articles was January 2018 to December 2023.

As in Van Breda and Theron [27], the search terms were: 'resilien*' and 'South Africa' and (1) 'child*', or (2) 'youth', or (3) 'young' or (4) 'adolescen*' or (5) 'teen*' or (6) 'learner*' or (7) 'student*'. The search terms were applied to titles, abstracts, and keywords/subject terms.

In total, the search yielded 1309 records. Fig 1 shows the PRISMA Flowchart depicting the search and selection process. Using Endnote software, the research psychologist identified and removed duplicates ($n = 745$) and ineligible records ($n = 271$) and transferred the remaining records ($n = 293$) to Rayyan. We (the authors) then independently screened these records to determine eligibility for full-text reading (i.e., we applied the eligibility criteria to each record). Following Saldaña [31], we met to reach consensus on the limited discrepancies in judgement ($n = 33$, 11% of the records). Ultimately, we selected 125 records for full-text reading and were able to retrieve the full texts of all 125 reports.

## Selection of sources of evidence

We each independently read 50% of the full texts to confirm their fit with the specified eligibility criteria, before moderating each other's decisions. Following a second consensus discussion to resolve the discrepancies in our assessments ($n = 18$; 14% of the full texts), we excluded 26 full texts (Fig 1), yielding 99 studies for the review.

## Data charting process

We used the same data charting form that informed the resilience review conducted by Theron, Cockcroft [32], which was an expanded version of Van Breda and Theron [27]. It included the study's aim, design, sample (size and specifics), risk exposure, better-than-expected outcomes and how they were measured/investigated, and PPFPs that were associated with better-than-expected outcomes. We each extracted data from 50% of the included full texts, before reviewing the other's extracted data.

## Collating, summarizing and reporting the results

Informed by guidance of qualitative content analysis in scoping reviews [33], we conducted a narrative synthesis of the data. We tabulated essential aspects of the included studies, including their design, the risk/s that participants were

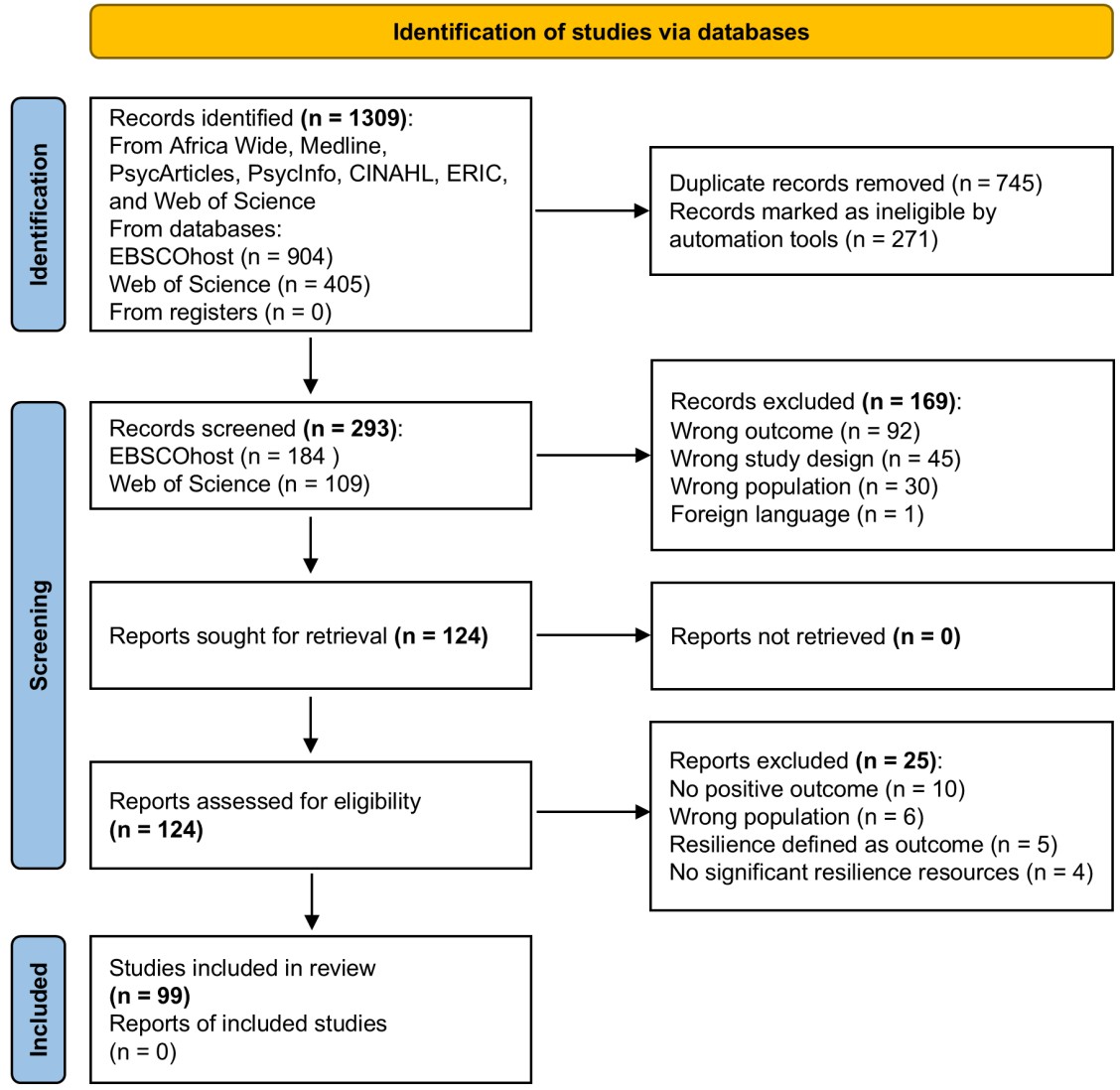

**Fig 1. PRISMA flow chart.**

exposed to, and better-than-expected outcome/s reported. We focused on the PPFPs associated with these outcomes. In line with a multisystemic approach to resilience [21–23,34], we considered the systems represented by the PPFPs, how (if at all) they represented multiple systems, and how (if at all) the PPFPs interacted. Our identification of systems was guided by the definitions summarized in Table 1.

## Results

We report two sets of results. Building on the data extraction table in S1 Table, which summarizes the key details of the 99 full texts included for review, we first provide an overview of the included studies (i.e., we comment on patterns relating to methodology, participants, the risks framing the studies, and better-than-expected outcomes). Thereafter, we report the PPFPs associated with the resilience of South African children and youth and comment critically on their alignment with multisystemic perspectives of resilience.

**Table 1. Systemic definitions informing how results were reported.**

| System | Definition |
|---|---|
| Biological | Everything about the physical body |
| Psychological | Everything related to feelings, thoughts, and behaviors (including interactions initiated by the individual) |
| Social | Everything related to relationships (i.e., connections with other people, animals, or beings that provide any form of social [informational, material, affective] support) |
| Institutional | Everything related to organizations, services, and policies |
| Structural | Everything related to economic standing (e.g., food security, level of education, employment), or the structure of a household (e.g., single-parent, orphan status, multi-generational) or community (e.g., law-abiding community, low exposure to violence) |
| Environmental | Everything related to the built or natural environment (e.g., quality housing, accessible green spaces) |
| Cultural | Everything related to beliefs, values and traditional/shared practices (e.g., religion, Ubuntu values) |

## Overview of the included studies

**Overview of designs.** A quantitative design informed half (*n* = 48) of the included studies. Qualitative designs were almost as popular (*n* = 43), with just over a third reporting visual methodologies (*n* = 16) [35–50]. Mixed methods designs were rare (*n* = 8) [44,46,51–56]. Across all these designs, longitudinal studies were infrequent (*n* = 12) [44,47,56–65]. Of these 12 studies, seven had just two waves of data, while two had three waves [44,57], and one each had five [63], six [59] or eight waves [58] of data. Eight of the longitudinal studies were quantitative, two were qualitative [47,63] and two mixed methods [44,56].

**Overview of participants.** The included studies represented a total of 57,411 participants. This total excludes repeated samples (i.e., those that were identically reported in multiple papers, albeit for the purpose of different analyses): [52] and [53]; [66] and [67]; [68,69], and [70]; and [63] and [71]. The total also excludes one study that did not report an exact sample size [72].

Most participants were adolescents (10- to 17-year-olds) or emerging adults (18- to 29-year-olds), with 56 studies including either adolescents or emerging adults and 28 studies including a mix of the two (e.g., 14- to 24-year-olds). Only 14 studies included infants and children (up to age 9). Most samples included young people engaged in education (primary, secondary, or tertiary level).

Seventeen studies focused on the resilience of girls or young women. Typically, they foregrounded gendered risks, including intimate partner violence [73,74], living with communicable disease [75], teenage motherhood [54,76–79], and sexual and reproductive health [66,67]. In contrast, only two studies focused on the resilience of young men. One explored resilience to school failure [80], and the other the resilience of teenage fathers [81]. Despite the relatively high number of girl-focused studies, only one [82] provided a gendered account of young women's resilience. Overall, most studies (*n* = 78) were not gender specific. Of these, 64 studies reported male-female participant ratios, with girls and young women representing 55% of the sample on average.

**Overview of risk contexts.** Almost half the studies (*n* = 44) investigated child or youth resilience in the context of structural violence (e.g., exposure to poverty, poor service delivery, overcrowding, physical and sexual violence, and marginalization) and/or family adversity. There was also repeated attention to risks that intersect with structural violence, including young people living with or affected by HIV (*n* = 11) [37,40,57,66,75,83–88]; transitioning to or progressing through university (particularly among first-generation students) (*n* = 9) [55,89–96]; living in or transitioning out of care (*n* = 7) [38,41,58,82,97–99]; and teenage parenthood (*n* = 6) [54,76–79,81].

Unsurprisingly, given the review period (2018–2023), many studies investigated the resilience of young people to challenges associated with the COVID-19 pandemic, such as lockdown and school closures (*n* = 17) [39,41,49,52,53,63–65,68–71,91,100–103].

The inattention to climate emergency and NEEThood risks was conspicuous. Apart from Theron, Mampane's [45] drought-focused study, no studies reported South African youth resilience to the climate emergency. Similarly, while there was passing mention of disengagement from education in the studies that explored the resilience of young mothers who returned to school [54,76–79], no study investigated the resilience of young people who were NEET.

**Overview of better-than-expected outcomes.** A third of studies (*n* = 33) conceptualized mental health (e.g., no or minimal symptoms of depression) or well-being (e.g., satisfaction with life) as the better-than-expected outcome of choice (or one of the outcomes of choice), and typically used validated scales (e.g., the Beck Depression Inventory-II or Strengths and Difficulties Questionnaire) to measure this outcome. Another third (*n* = 29) used academic markers to denote the better-than-expected outcome. This included academic persistence (e.g., returning to school after becoming a teenage parent or not quitting the first year of university), academic success or progress (e.g., passing a semester or being in the age-appropriate grade), or engagement in schooling. These outcomes were measured (e.g., using academic grades or records, or validated scales, such as the School Engagement Scale [104]), or self- or other-reported (e.g., by teaching staff). A fifth of the studies used a behavioral or health outcome (*n* = 18; e.g., engagement in leisure activity or non-engagement in risky sexual behaviors), positive coping, often broadly described as 'doing well' and typically self-reported (*n* = 14); and/or a developmental outcome (*n* = 9; e.g., independent living among young adults or normative cognitive development among pre-school children).

## PPFPs by system

**Frequency of systems.** As in Van Breda and Theron [27], resources in the psychological system are the most frequently reported in the 2018–2023 papers included in this review (*n* = 75), followed by the social system (*n* = 68). These two systems are by far the most frequently reported resilience-enabling systems. After these, 27 studies reported PPFPs in the institutional system, followed by 20 in the cultural, 13 in the structural, nine in the environmental and eight in the biological systems. The comparative under-reporting of institutional, cultural, structural, environmental, and biological resources mimics the resource pattern reported by Van Breda and Theron [27]. Then and now, it is the most micro system (biological) and the most macro systems (structural, environmental, and cultural) that are the least frequently reported, while the three in-between systems (psychological, social, and institutional) are most frequently reported.

**Multiple systems of resilience resources.** Two-system combinations trump monosystem and multisystem accounts. As shown in Table 2, one-third (*n* = 31) of the 99 studies identified resilience resources in only one system. Of these, half (*n* = 15) were in the psychological system, a third (*n* = 10) in the social system, three institutional, two cultural, and one structural. This appears to confirm a trend towards mono-systemic and psychosocial constructions of resilience, rather than multisystemic constructions.

However, this does mean that around two-thirds of the studies (*n* = 68) incorporated multiple systems. Almost half of these (*n* = 31) identified resilience resources in two systems. About two-thirds of these (*n* = 19 of 31) were a combination of the psychological and social systems. Other two-system combinations included three studies combining psychological and cultural systems, and three combining social and structural systems. The remaining six studies with two-system combinations each had a different pairing of systems (i.e., Biological + Personal; Biological + Structural; Psychological + Institutional; Psychological + Structural; Social + Institutional; and Institutional + Structural).

One quarter (*n* = 23) of the 99 studies had three-system combinations, most of which (*n* = 21) are based on the familiar psychosocial combination, plus a third system. Ten of these included the institutional system, five the cultural system, three structural, two environmental, and one biological. The remaining two studies combined Biological + Social + Institutional and Social + Structural + Environmental.

**Table 2. Summary of monosystems and multisystems of PPFPs.**

| System count | System combinations | n | Studies |
|---|---|---|---|
| 1 | Psychological | 15 | [66,68–70,73,79,94–96,99,102,105–108] |
| 1 | Social | 10 | [42,48,57,101,109–114] |
| 1 | Institutional | 3 | [61,72,98] |
| 1 | Structural | 1 | [115] |
| 1 | Cultural | 2 | [116,117] |
| 2 | Psychological+Social | 19 | [36,38,41,50–52,67,71,75,77,78,80,84,88,89,103,118–120] |
| 2 | Psychological+Cultural | 3 | [82,121,122] |
| 2 | Social+Structural | 3 | [86,87,123] |
| 2 | Biological+Personal | 1 | [93] |
| 2 | Biological+Structural | 1 | [124] |
| 2 | Psychological+Institutional | 1 | [64] |
| 2 | Psychological+Structural | 1 | [125] |
| 2 | Social+Institutional | 1 | [54] |
| 2 | Institutional+Structural | 1 | [126] |
| 3 | Psychological+Social+Institutional | 10 | [35,45,55,58,65,90,91,127–129] |
| 3 | Psychological+Social+Cultural | 5 | [47,76,97,130,131] |
| 3 | Psychological+Social+Structural | 3 | [39,60,92] |
| 3 | Psychological+Social+Environmental | 2 | [46,49] |
| 3 | Biological+Psychological+Social | 1 | [100] |
| 3 | Biological+Social+Institutional | 1 | [56] |
| 3 | Social+Structural+Environmental | 1 | [59] |
| 4 | Psychological+Social+Institutional+Cultural | 5 | [40,43,62,83,132] |
| 4 | Psychological+Social+Structural+Cultural | 2 | [74,81] |
| 4 | Biological+Psychological+Social+Structural | 1 | [85] |
| 4 | Biological+Psychological+Social+Environmental | 1 | [133] |
| 4 | Biological+Psychological+Institutional+Structural | 1 | [134] |
| 4 | Psychological+Social+Institutional+Environmental | 1 | [53] |
| 4 | Psychological+Social+Environmental+Cultural | 1 | [44] |
| 5 | Psychological+Social+Institutional+Environmental+Cultural | 2 | [37,63] |

Twelve studies had four-system combinations, 11 of which were based on the psychosocial combination plus two other systems. Five of these were combined with institutional and cultural systems, and two with structural and cultural systems. The remaining five studies each had a unique combination of systems, three of which were based on the psychosocial pair. These four-system studies, while still based on the psychosocial pair, show a diverse engagement with a wide range of other systems – each of the systems was referenced in at least three of the 12 studies.

Finally, two studies had a five-system combination, both with the same systems: the ubiquitous psychosocial pair, plus institutional, environmental, and cultural systems. Neither study drew on the biological or structural systems.

Overall, 44 of the 99 studies focused on psychological or social systems or both, confirming a focus on the traditional psychosocial domains of resilience, regardless of the number of systems included (i.e., mono, two, or multiple).

## Co-acting systems

Multisystemic resilience involves not only the presence of multiple PPFPs, but also the co-acting of these systems with each other. In other words, the contribution of multiple PPFPs to better-than-expected outcomes is enhanced when these systems interact or synergize with one another. Establishing co-action between PPFPs is a recent innovation and the methodologies for detecting such co-actions are still evolving [5]. Most studies did not report on co-action between systems, but rather on the contributions of separate systems. The subsequent sections provide examples of studies that did report co-action.

**Qualitative findings.** In some qualitative studies, accounts of the interactions between systems are reported explicitly or implied. For example, Bezuidenhout, Theron [36] described the interactions between psychological and social systems in a grade 1 child. The child is described by her mother as "a driven child". The child shares with the interviewer her excitement at getting high scores for excellent performance. The teacher acknowledges this and affirms the child with positive feedback. Furthermore, the child also relates building networks of friends, who help moderate her performance anxiety, based on a larger anxiety concerning her parents' divorce. The teacher also assisted in containing the child by giving her a hug. The child's sense of self, combined with self-initiated and other-initiated support from peers and a kind teacher, fostered co-action between the psychological and social systems.

In another qualitative study, Bond and Van Breda [38] suggested interactions between children's possible selves and turning point people or role models in their social environment. One participant reported being recruited into a gang, involving drug use. In his community, gang membership can be difficult to withstand, but joining a gang also meets powerful social needs for belonging, identity, and protection [what Pessoa et al. [135] refer to as 'hidden resilience']. Around this time, a pilot visited the residential care facility where the child was living and asked him what he wanted to be and encouraged him to consider becoming a pilot. The child was impressed and inspired, and cultivated a hoped-for possible self as an air technician, which helped to decenter the pressure from the gang. The possible self was a psychological construct, enabled by a natural mentor, that loosened the power of the gang. Here, two social systems influenced and were in turn influenced by the psychological system, co-acting with each other.

A single qualitative study explicated co-acting PPFPs. Theron, Murphy [133] identified several systemic interactions in their thematic account of the resilience of 21 youth from a stressed community in South Africa. The reported interactions included PPFPs from two, three and four systems co-acting to support youth resilience (i.e., psychological + informal social resources; biological + psychological + informal social resources; and biological + psychological + social + built environment resources).

**Quantitative findings.** In a few quantitative studies, statistical methods were used to explore the interactions between systems. In a study of predictors of school engagement [56], age (in the biological system), parental or caregiver warmth (social system), school resources (institutional system) and perceived teacher competence (institutional system) co-acted in complex ways. For children under 16 years, school engagement was predicted by a well-resourced school, and significantly enhanced by high levels of parental warmth; while for children over 16 years, high parental warmth was highly predictive of school engagement, and further strengthened by perceiving their teacher to be competent. This analysis shows the unique ways that three systems (biological, social and institutional) co-act to influence school engagement.

Another study [44] considered the contribution of multiple systems of resilience to depression among 15–23-year-old youth in high-risk communities. The researchers found that psychological systems (e.g., hopefulness and future orientation), social systems (e.g., family, peer, and school), environmental (e.g., sports facilities) and cultural (e.g., faith-based beliefs and practices) were present among all participants in the study, regardless of the depression trajectory reported over an 18-month period (i.e., chronic high depression, worsening depression, declining depression, or stable

low depression). However, those with a declining or stable low depression trajectory over time reported more varied and deeper combinations of these systems than those with higher or worsening depression trajectories.

Other differences included how these systems were utilized. For example, most participants turned to cultural, specifically faith-based practices. Those with declining depression included faith-based practices and did so frequently over time, while those with worsening depression were more likely to include faith-based practices, but did so less frequently over time. In addition, those with lower or lessening depression reported greater use of environmental systems like libraries and parks, and to draw on psychological resources like future orientation. The researchers concluded that multiple interacting systems, particularly environmental and cultural systems, facilitate decreasing or lower depression in contexts of high adversity.

## Discussion

Drawing on a narrative synthesis of 99 eligible articles published between 2018 and 2023, this scoping review updates the Van Breda and Theron [27] review of child and youth resilience studies conducted in South Africa. The update focuses on identifying the PPFPs associated with the resilience of South African young people, with emphasis on how these align with more recent multisystemic approaches to resilience. Multisystemic approaches theorize that co-acting PPFPs, found across biological, psychological, social, institutional, structural, environmental, and cultural systems, scaffold better-than-expected child and youth outcomes [4,21–23]. A better understanding of the multisystemic roots of resilience is pivotal to optimizing resilience-enabling supports to young people challenged by significant risk, as well as to designing future resilience studies that halt the reification of simplistic (i.e., mono-systemic) accounts of child and youth resilience.

As in the Van Breda and Theron [27] review, psychological and social PPFPs dominated accounts of the resources supporting child and youth resilience in the 2018–2023 period. While this joint focus is an encouraging move away from the traditional and inadequate preoccupation with only psychological strengths when explaining child and youth resilience [23], it still eclipses the potential contribution of biological, institutional, structural, environmental, and cultural systems. Studies of how best to accelerate positive SDG-aligned child and youth outcomes clearly show the value of not limiting supports to psychological and social ones [136]. For example, lower exposure to abuse for adolescents living with HIV in the Eastern Cape in South Africa was associated not only with social supports (i.e., better parenting), but institutional (i.e., household cash transfers) and environmental (i.e., safe school spaces) supports too [137]. Similarly, school-attending adolescents in Nigeria reported lower substance use when they were mentally healthy, attended schools that were safe, and experienced food security (i.e., a combination of psychological, environmental, and institutional PPFPs) [138].

Two-thirds of the studies ($n = 68$) reported in this review included multiple systems. While this suggests allegiance with the recent trend toward multisystemic accounts of resilience, enthusiasm for this finding is tempered by the fact that about half ($n = 31$) focused on only two systems, with most two-system studies focusing on psychological and social systems. This narrow focus could be an artefact of how challenging it is to design resilience studies that involve three or more systems. For example, researchers need to determine which systems to include and how contextually and culturally relevant they are to a given population; recruit a large enough sample to facilitate an adequately powered analysis of multiple variables; and identify analysis strategies that can combine diverse systemic data [5]. Further, working across systems implies navigating the complexities of collaboration with researchers from multiple disciplines (i.e., researchers who do not necessarily share common theoretical frameworks, scientific jargon or research methodologies).

Still, despite the popularity of psychosocial systems, the remaining multisystemic studies ($n = 37$) broadened the focus on psychological and social PPFPs to include PPFPs from biological, institutional, structural, environmental and cultural systems. Among these, biological and environmental systems were least represented. In addition to the above-mentioned challenges, the under-representation of these systems could be an artefact of gaining youth (and caregiver) buy-in to provide biological samples [e.g., 139], the relative absence of safe green spaces in Southern Africa's resource-constrained

communities [140], and the expense of objectively measuring biological and environmental PPFPs. Investigating psychosocial and related PPFPs (e.g., cultural or institutional PPFPs) is less costly, not least because it can be done via self-report.

Although the attention to multiple systems is a positive development in the studies of South African child and youth resilience, more attention is needed to how the PPFPs in diverse systems co-support better-than-expected outcomes. Identifying resources across systems that support child and youth resilience falls short of accounting for their co-occurring protective effects. Investigating co-acting PPFPs is key to discerning which combinations of resources matter most for which better-than-expected outcomes for which children and youth in what contexts of risk [1,5,29], and for determining additive protective effects [136]. To identify co-acting resources, future quantitative studies need to adopt sophisticated multilevel modelling approaches, such as network analyses [4,141]. Similarly, research questions that enquire about interacting PPFPs during data generation or analytic questions that purposefully identify co-acting PPFPs during data analysis can provide useful insight into resource combinations and their protective value.

These findings, specific to South Africa, share similarities with the state of resilience research in other countries, not only in the majority world, but also the minority world. For example, resilience scholars in Canada note the emphasis on individual coping strategies and the use of only one or two systems incorporated in resilience studies [139]. As recently as 2024, Masten, based in the US, describes multisystemic resilience as still emerging and possible [1], suggesting that these South African findings have global relevance.

## Limitations

We report that almost a third of the included studies were mono-systemic studies. However, there were instances where studies investigated resources in multiple systems, but only a single resource was found to be significantly protective [e.g., 108]. Had we reported non-significant results, the number of multisystemic studies would have been higher.

As documented in the overview of the included studies, children younger than nine were poorly represented. Similarly, although there are growing numbers of South African youth with experience of NEEThood and concerns that the climate emergency will swell these numbers [17], there was almost no attention to the resilience of young people exposed to either NEEThood or extreme weather events. The limited representation of these children and youth cautions against assumptions that the review results are applicable to them.

Also, only a minority of studies were gender specific and of these, only one [82] provided a gendered account of resilience. Following [142], the inattention to gendered processes of resilience is a limitation that should be rectified by future resilience studies.

## Conclusion

Taken together, the results of this scoping review show that 2018–2023 studies of South African child and youth resilience have not been agnostic to a multisystemic approach to resilience. A multisystemic approach is essential to gaining a holistic understanding of the resources that should be put in place to support better-than-expected outcomes among vulnerable children and youth. In contexts like South Africa, where structural violence is putatively intractable, a multisystemic approach urges government and other stakeholders to replace the neoliberal rhetoric of resilience with policies and action that make systemic resources available to those who are vulnerable [2]. Facilitating biological, institutional, structural, environmental, and cultural resources should relieve vulnerable young people and their support networks from shouldering the responsibility for child and youth resilience.

Going forward, however, researchers will need to account for how biological, psychological, social, institutional, structural, environmental, and cultural resources combine to co-facilitate better-than-expected outcomes among South Africa's many vulnerable children and youth. Ideally, they need to do so over time, given the dynamic nature of resilience [1,5]. These future studies also need to redress the under-representation of some of South Africa's most vulnerable children

and youth in resilience studies to date (e.g., those with experience of extreme weather events, climate change impacts or NEEThood).

## Supporting information

**S1 Checklist. PRISMA 2020 checklist.**
(DOCX)

**S1 Table. Profile of South African child and youth resilience studies, 2018–2023.**
(DOCX)

## Author contributions

**Conceptualization:** Linda C. Theron, Adrian D. van Breda.

**Data curation:** Linda C. Theron, Adrian D. van Breda.

**Formal analysis:** Linda C. Theron, Adrian D. van Breda.

**Methodology:** Linda C. Theron, Adrian D. van Breda.

**Writing – original draft:** Linda C. Theron, Adrian D. van Breda.

**Writing – review & editing:** Linda C. Theron, Adrian D. van Breda.

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
