## [Editor Report · Decision Letter 0]

26 Dec 2024

Dear Dr. van Breda,

Thank you for submitting your manuscript to PLOS ONE. After careful consideration, we feel that it has merit but does not fully meet PLOS ONE’s publication criteria as it currently stands. Therefore, we invite you to submit a revised version of the manuscript that addresses the points raised during the review process.

We look forward to receiving your revised manuscript.

Kind regards,

Azizollah Arbabisarjou, Ph.D

Academic Editor

PLOS ONE

Journal Requirements:

Additional Editor Comments :

The article is too long paper . the methods are nice and clear. It suggested to decline the words to 10000 and the references to 55-60

he whole paper must be re-edit and revise by English editor ( professional)
---

## [Author Response · Author response to Decision Letter 1]

2 Mar 2025

Dear Editor, thank you for your feedback on our manuscript. We have made the requested revisions. These are set out below:

1. We have checked and corrected conformity to the journal’s manuscript body formatting guidelines

2. We have checked the cover page conformity to the title, author, affiliations formatting guidelines. We have removed our ORCID numbers as the are not mentioned in the guidelines. We have also removed the & from the second author and marked both with ¶, since both authors contributed equally to the paper.

3. We have corrected the labelling of Figures.

4. We have corrected the font size of Level 2 headings to 16pt and Level 3 to 14pt.

5. We have corrected the font size of table headings to 12pt.

6. To reduce the length of the manuscript, we have moved Table 2 into a supplementary file (S1 Table). This brings the manuscript to within 10,000 words as requested.

7. We have aligned Table 1 to the left, rather than centred, for easier reading.

8. We have ensured that all tables and figures appear immediately after they are first referred to in text.

9. We have carefully proofread the manuscript and corrected language errors.

10. We identified a missing space after the publication dates in the reference list and have corrected this.

11. We have checked all manuscripts to ensure they have not been retracted and to make any necessary updates. The following references were changed:

a. [1] Masten – this advance online publication has since been published

b. [25] Theron – this advance online publication has since been published

c. [138] Tamambang – this advance online publication has since been published

12. Two papers in the systematic review were advance online publications when we conducted the search, and have since been published. We have retained the references at the time of the search:

a. [106] Crombie KE, Crombie KD, Salie M, Seedat S. Medical students’ experiences of mistreatment by clinicians and academics at a South African university. Teaching and Learning in Medicine. 2023:Advance Online Publication. doi: 10.1080/10401334.2023.2167207

b. [58] Van Breda AD. The contribution of supportive relationships to care-leaving outcomes: A longitudinal resilience study in South Africa. Child Care in Practice. 2022:Advance online publication. doi: 10.1080/13575279.2022.2037516

13. One reference is that is part of the article text, not the review, is still an advance online publication. We have retained its formatting:

a. [106] Crombie KE, Crombie KD, Salie M, Seedat S. Medical students’ experiences of mistreatment by clinicians and academics at a South African university. Teaching and Learning in Medicine. 2023:Advance Online Publication. doi: 10.1080/10401334.2023.2167207

14. We do recognise that the reference list is substantial. However, we wish to note that 99 of the references are the texts that comprises the scoping review itself. We cannot remove any of these, as these are our core evidence. The remaining 44 references are, in our view, necessary, as the provide the contextual framework for the study overall and particularly for the multisystemic approach to resilience, which, being a recent innovation on resilience science, requires sufficient citation to the scholarship of the field. We thus believe that removing references will harm the integrity and scholarliness of the paper.

---

## [Decision Letter · Decision Letter 1]

29 Oct 2025

The multisystemic roots of South African child and youth resilience: A scoping review

PONE-D-24-54938R1

Dear Dr. Van Breda,

We’re pleased to inform you that your manuscript has been judged scientifically suitable for publication and will be formally accepted for publication once it meets all outstanding technical requirements.

Kind regards,

AKM Alamgir, PhD

Academic Editor

PLOS ONE

Additional Editor Comments (optional):

Reviewers' comments:

Reviewer's Responses to Questions

**Comments to the Author**

Reviewer #1: (No Response)

2. Is the manuscript technically sound, and do the data support the conclusions?

Reviewer #1: Yes

3. Has the statistical analysis been performed appropriately and rigorously?

Reviewer #1: N/A

4. Have the authors made all data underlying the findings in their manuscript fully available?

Reviewer #1: Yes

5. Is the manuscript presented in an intelligible fashion and written in standard English?

Reviewer #1: Yes

Reviewer #1: (No Response)

**Do you want your identity to be public for this peer review?** For information about this choice, including consent withdrawal, please see our Privacy Policy

Reviewer #1: No

---

## [Editor Report · Acceptance letter]

PONE-D-24-54938R1

PLOS ONE

Dear Dr. van Breda,

I'm pleased to inform you that your manuscript has been deemed suitable for publication in PLOS ONE. Congratulations! Your manuscript is now being handed over to our production team.

Kind regards,

on behalf of

Dr AKM Alamgir

Academic Editor

PLOS ONE